# Long-term mortality outcome of a primary care-based mobile health intervention for stroke management: Six-year follow-up of a cluster-randomized controlled trial

Xingxing Chen[1,2©], Enying Gong[3,4©*], Jie Tan[1,2©], Elizabeth L. Turner[5,6], John A. Gallis[5,6], Shifeng Sun[5,6], Siran Luo[2], Fei Wu[7], Bolu Yang[1,2], Yutong Long[3], Yilong Wang[8], Zixiao Li[8], Yun Zhou[8], Shenglan Tang[2,5], Janet P. Bettger[5], Brian Oldenburg[9,10], Xiaochen Zhang[2], Jianfeng Gao[1], Brian S. Mittman[11], Valery L. Feigin[12], Ruitai Shao[3,4], Shah Ebrahim[13], Lijing L. Yan[1,2,5*]

1 School of Public Health, Wuhan University, Wuhan, China, 2 Global Health Research Center, Duke Kunshan University, Suzhou, China, 3 School of Population Medicine and Public Health, Chinese Academy of Medical Sciences and Peking Union Medical College, Beijing, China, 4 State Key Laboratory of Respiratory Health and Multimorbidity, Chinese Academy of Medical Sciences and Peking Union Medical College, Beijing, China, 5 Duke Global Health Institute, Duke University, Durham, North Carolina, United States of America, 6 Department of Biostatistics and Bioinformatics, Duke University, Durham, North Carolina, United States of America, 7 Department of Global Health and Population, Harvard T.H. Chan School of Public Health, Boston, Massachusetts, United States of America, 8 Beijing Tiantan Hospital, Capital Medical University, Beijing, China, 9 Baker Heart and Diabetes Institute, Melbourne, Australia, 10 La Trobe University, Melbourne, Australia, 11 Department of Research and Evaluation, Kaiser Permanente, Pasadena, California, United States of America, 12 National Institute for Stroke and Applied Neurosciences, Auckland University of Technology, Auckland, New Zealand, 13 London School of Hygiene & Tropical Medicine, London, United Kingdom

© These authors contributed equally to this work and should be considered as co-first authors.
* gongenying@cams.cn (EG), lijing.yan@duke.edu (LLY)

## Abstract

### Background

Despite growing evidence of primary care-based interventions for chronic disease management in resource-limited settings, long-term post-trial effects remain inconclusive. We investigated the association of a 12-month system-integrated technology-enabled model of care (SINEMA) intervention with mortality outcomes among patients experiencing stroke at 6-year post-trial.

### Methods and findings

This study (clinicltiral.gov registration number: NCT05792618) is a long-term passive observational follow-up of participants and their spouse of the SINEMA trial (clinical-trial.gov registration number: NCT03185858). The original SINEMA trial was a cluster-randomized controlled trial conducted in 50 villages (clusters) in rural China among patients experiencing stroke during July 2017–July 2018. Village doctors in the intervention arm received training, incentives, and a customized mobile health application supporting monthly follow-ups to participants who also received daily free automated

**Data availability statement:** No patient-level data collected in this trial can be made available externally owing to ethical regulations and data regulations. Upon reasonable request, non-identifiable aggregated data could be shared pending approval from the steering committee and from the institutional review boards. For inquiries regarding data usage or code availability, please contact the email of the manager of our database bingyi.wang@ dukekunshan.edu.cn.

**Funding:** The post-trial follow-up of SINEMA was funded by the National Natural Science Foundation of China (Grant No. HH8220122034 to EG), the non-profit Central Research Institute Fund of Chinese Academy of Medical Sciences (Grant No. 2021-RC300-004 to EG), and the National Key R&D Program of China (Grant No. 2023YFC3605000 to LLY). The SINEMA initiative was funded by the United Kingdom Medical Research Council, Economic and Social Research Council, Department for International Development, and Wellcome Trust (Grant No. MR/N015967/1 to LLY). The research results of this article are sponsored by the Kunshan Municipal Government research funding (LLY). The funders had no role in study design, data collection and analysis, decision to publish, or preparation of the manuscript.

**Competing interests:** The authors have declared that no competing interests exist.

**Abbreviations :** CDC, Center for Disease Control and Prevention; cRCTs, cluster-randomized controlled trials; HRs, hazard ratios; LMICs, low- and middle-income countries; SINEMA, system-integrated mobile-health technology-enabled model of care; 95% CIs, 95% confidence intervals.

voice-messages. Vital status and causes of death were ascertained using local death registry, standardized village doctor records, and verbal autopsy. The post-trial observational follow-up spanned from 13- to 70-months post-baseline (up to April 30, 2023), during which no intervention was requested or supported. The primary outcome of this study was all-cause mortality, with cardiovascular and stroke cause-specific mortality also reported. Cox proportional hazards models with cluster-robust standard errors were used to compute hazard ratios (HRs) and 95% confidence intervals (95% CIs), adjusting for town, age, and sex in the main analysis model. Analyses were conducted on an intention-to-treat basis.

Of 1,299 patients experiencing stroke (mean age 65.7 years, 42.6% females) followed-up to 6 years, 276 (21.2%) died (median time-to-death 43.0 months [quantile 1–quantile 3: 26.7–56.8]). Cumulative incidence of all-cause mortality was 19.0% (121 among 637) in the intervention arm versus 23.4% (155 among 662) in the control arm (HR 0.73; 95% CI 0.59, 0.90; $p = 0.004$); 14.4% versus 17.7% (HR 0.73; 95% CI 0.58, 0.94; $p = 0.013$) for cardiovascular cause-specific mortality; and 6.0% versus 7.9% (HR 0.71; 95% CI 0.44, 1.15; $p = 0.16$) for stroke cause-specific mortality. Although multi-source verification was used to verify the outcomes, limitations exist as the survey- and record-matching-based nature of the study, unavailability of accurate clinical diagnostic records for some cases and the potential confounders that may influence the observed association on mortality.

## Conclusions

Despite no observed statistically difference on stroke cause-specific mortality, the 12-month SINEMA intervention, compared with usual care, significantly associated with reduced all-cause and cardiovascular cause-specific mortality during 6 years of follow-up, suggesting potential sustained long-term benefits to patients experiencing stroke.

## Author summary

### Why was this study done?

- Trials have confirmed the within-trial effectiveness of primary care-based or community-based interventions for stroke management, while effectiveness of such interventions integrated with mobile health technology were inconclusive.

- Most primary care-based trials typically focused on cardiovascular risk factors as primary outcomes, with results assessed immediately after the intervention phase was completed.

- Evidence is lacking regarding the long-term post-trial effectiveness of such interventions, particularly on mortality.

### What did the researchers do and find?

- We conducted a 6x-year long-term follow-up of a primary care-based cluster-randomized controlled trial and ascertained the survival status of all enrolled participants.

- Six-year after the 12-month primary care-based mobile health intervention, we observed a reduction in all-cause and cardiovascular cause-specific mortality among patients

with stroke in rural China, which may indicate sustained long-term benefits of the intervention.

- Stronger associations were observed among more vulnerable individuals, which may indicate the inequity-reducing impact of the intervention.

## What do these findings mean?

- Our results extend previous within-trial evidence on the effectiveness of primary care-based strategies to promote secondary prevention of stroke in resource-limited settings by demonstrating a potential long-term association with mortality, even when the intervention was not requested further.

- Although the generalizability of the findings may be limited due to the single study site, there is a potential for adapting and scaling-up the SINEMA model to other chronic diseases in China and beyond.

## Introduction

Stroke is one of the leading causes of premature mortality and significant disability, particularly in low- and middle-income countries (LMICs), where more than 85% of stroke-related fatalities and 89% of disability-adjusted life years occur globally [1]. Primary care-based stroke management for community-dwelling patients such as regular visits by primary care providers with therapeutic lifestyle recommendations is a pragmatic and promising strategy to improve patient outcomes. A limited number of cluster-randomized controlled trials (cRCTs) conducted in LMICs have demonstrated the benefit of such interventions for stroke management [2–4]. Typically, these trials have only evaluated short-term effectiveness for cardiovascular risk factors. Their long-term benefits beyond the active intervention phase on endpoints such as mortality remain unclear.

Evidence regarding long-term benefits on hard endpoints has crucial implications for sound policy making and improving population health. From 2017 to 2018, we conducted a 12-month cRCT of a primary-care based system-integrated mobile-health technology-enabled model of care (SINEMA) for stroke management in rural China [5]. This model strengthened the existing primary care workforce through training and support embedded in the entire healthcare system and innovatively integrated both provider-side and patient-facing mobile health (mHealth) technology. Compared with usual care, the intervention significantly reduced the primary outcome of systolic blood pressure at 12-month when the intervention ended [4].

We conducted post-trial assessment and followed up all participants until April 30, 2023 with ascertainment of vital status and causes of death. In this paper, we report the findings on long-term outcomes of all-cause and cause-specific mortality beyond the active intervention phase of participants post SINEMA trial.

## Methods

### Study design and participants

This study is part of the Stroke Patients and Family Longitudinal Study in Rural China (SaFaRI) study (clinicaltrial.gov registration number: NCT05792618), a long-term passive (observational) follow-up of participants and their spouse in the SINEMA cRCT (S1 Protocol). The original SINEMA trial (clinicaltrial.gov: NCT03185858) was designed to evaluate a primary care-based mobile health intervention for stroke management within 1 year with protocol published previously [5].

The trial was conducted in Nanhe County, Hebei, China, where stroke prevalence was twice the national average [6]. The trial included five out of eight townships, with 10 villages (clusters) recruited from each, totaling of 50 villages. Eligible villages had a minimum population size of 1,500 and at least one willing village doctor. From June 23 to July 21, 2017, we recruited 1,299 clinically stable patients experiencing stroke with confirmed diagnoses from county-level or higher hospitals. Exclusions were patients unable to get out of bed, those with severe life-threatening diseases, or an expected life span shorter than 6 months. Villages, along with their village doctors and recruited participants, were randomized in a 1:1 ratio to either the intervention ($n = 25$) or control arms ($n = 25$) by a biostatistician in the United States after baseline data collection, using a computer-generated random numbering system with stratification by township. The intervention lasted 12 months from July 2017 to July 2018.

Vital status for all participants was ascertained and verified from multiple sources (details below). After ascertaining vital status for all participants, we conducted follow-up surveys for those who were still alive in 2022–23 (Fig A in S1 File). This post-trial follow-up study was approved by the ethical boards of the Chinese Academy of Medical Sciences (approval number: CAMS&PUMC-IEC-2022-062 and CAMS & PUMC-IEC-2024-047 for renewing approval) and Duke University (approval number: Pro00082130-AMD-9.1) and reported according to the Consolidated Standards of Reporting Trials (S1 Checklist). All study participants or relatives of deceased participants provided written informed consent.

## Interventions and implementation strategies

The details of the intervention design and development have been published in three papers [5,7,8]. In summary, the SINEMA intervention was developed based on existing evidence-based practice, emphasized monthly follow-ups, medication adherence (antiplatelet, statin, and antihypertensive medication), and appropriate physical activity for stroke secondary prevention and was tailored to rural China's context and built upon clinical guidelines for stroke management at grass-root levels [9]. Implementation strategies aimed to enhance the existing primary care workforce through training, support, and mHealth technology integration. Village doctors, who can prescribe essential medications [10], received 1.5-day of training, performance-based financial and non-financial incentives, and equipped with an Android smartphone with the SINEMA App installed to support the delivery of monthly follow-up and patients management [7,8]. Village doctors provided monthly follow-ups to patients experiencing stroke by following standardized procedures, including blood pressure monitoring, stroke symptom review, medication adherence checks, and therapeutic lifestyle recommendations. Patients also received daily voice messages with emphasis on physical activity and medication adherence, which were automatically dispatched every morning—recorded in the local dialect [7]. Control arm received usual care, necessitating proactive care-seeking on their part, while village doctors continued their standard clinical practices.

After the 12-month intervention, participants were observed naturally without further intervention required or supported by the research team, while village doctors in the intervention arm could optionally continue offering these services on top of usual care, without financial incentive or further project support.

## Data collection for vital status and causes of death

For all participants, vital status and causes of death were ascertained from multiple sources until April 30, 2023, according to the following procedures. First, participants were linked to the vital registration system from Nanhe Center for Disease Control and Prevention (CDC) using unique identification numbers, which provided immediate and underlying causes of death coded by the

International Statistical Classification of Diseases and Related Health Problems, 10th Revision (ICD-10). We used the underlying causes of death to reflect the root causes. Second, standard-ized records from village doctors were used to supplement vital registration data. Third, for the rare inconsistency in dates or causes of death between these two available sources, trained local CDC assessors conducted a verbal autopsy with knowledgeable relatives using the digital verbal autopsy instrument of Population Health Metrics Consortium Shortened Questionnaire adult module in the SmartVA system, validated in rural China [11]. Minimal migration in the study population ensured complete and reliable mortality data (both date and cause).

The follow-up period for vital status lasted from baseline to date of death or April 30, 2023. For six decedents with missing dates of death (2% of confirmed deaths), dates were imputed using median survival times, which were 8.69 months for five individuals who died during the 1-year trial phase and 46.39 months for one individual who died during the post-trial phase.

## Follow-up surveys among participants

Trained assessors from the CDC in a neighboring county conducted data collections for baseline, 12-month, and 6-year follow-up in October 2022 and May 2023. These assessors were not involved in any of the program implementation and were blinded to the intervention allocation. For all three time points, the same standard protocol was used for data collection in identical ways across all villages and for all participants. The survey was administered through face-to-face interviews with a web-based electronic data capture platform—Qualtrics (Provo, UT, version 10.2022).

## Outcomes

The primary outcome of this long-term follow-up study followed the original SINEMA trial in systolic blood pressure but also consider all-cause mortality as pre-specified long-term co-primary outcomes as described in the study protocol (S1 Protocol). The secondary outcomes include dia-stolic blood pressure, medication adherence (measured using the 4-item Morisky Green Levine Scale), physical activity level (measured by the short version of the International Physical Activity Questionnaire), mobility (measured using the Timed-Up and Go test), health-related quality of life (measured using the European Quality of Life 5 Dimensions 5 Level Version), disability (measured using the modified Rankin Scale), stroke recurrence (measured by self-reported expe-rience of stroke recurrence), stroke-related hospitalization (measured by self-reported history of hospitalization) and other comorbidities (such as cognition impairment).

This paper considered all-cause mortality as primary outcome. Causes of deaths were classi-fied by using ICD 10 codes based on the Global Burden of Diseases, Injuries, and Risk Factors Study into cardiovascular diseases (including stroke and non-stroke cardiovascular diseases) and other causes (Table A in S1 File) [12]. We also illustrate a subset of secondary outcomes as major intermediate outcomes of mortality, including changes in blood pressure, medication adherence to antiplatelet, statin and antihypertensive medications, changes in physical activity, function-ing, disability and stroke hospitalization measured through follow-up surveys. Definitions and measurement of these outcomes have been specified previously and in Appendix A in S1 File [5]. A detailed analysis in systolic blood pressure and blood pressure control related outcomes in medication use and stroke recurrence have recently been reported separately [13] and changes in cognition and mental health status will also be reported separately.

## Statistical analysis

Continuous variables were summarized as means and standard deviations, and categorical variables as counts and percentages. All analyses followed the intention-to-treat principle.

Mortality rates were estimated as events per 1,000 person-years since baseline. Cumulative incidence for mortality was graphed using the cumulative incidence function.

Cox proportional hazards model with cluster-robust standard errors were used to compute hazard ratios (HRs) and 95% confidence intervals (95% CIs) [14]. Models were fitted using the R "survival" package [15], and adherence to the proportional hazards assumption was tested with smoothed Schoenfeld residuals, showing no violations. For three cause-specific mortality outcomes (cardiovascular, the sub-type of stroke-specific, and non-cardiovascular causes), Cox models were used to estimate HRs, with participants censored at the time of a competing event. Age, sex, and township were adjusted for in the minimally adjusted model, while the fully adjusted model included these variables along with additional baseline variables noted to be significantly different between the arms at baseline ($P < 0.05$), including baseline diastolic blood pressure, having hypertension, having none of the assessed home assets (TV, refrigerator, air conditioner, and computer), and taking antihypertensive medicines [16]. Sensitivity analysis used the Fine and Gray approach for cause-specific mortality, accounting for competing risks using the sub-distribution model, with the same set of adjustments [17].

Subgroup analysis were performed by prespecified subgroups (age, sex, education, and duration since the first stroke event) used in the 12-month outcome analyses [16], and post-hoc subgroups (baseline status of annual income, marital status, systolic blood pressure, disability status, and stroke type). The interaction terms between the subgroup variables and the intervention were added as fixed effects.

We also analyzed intermediate outcomes among participants who completed the post-trial 6-year follow-up survey in 2022–23, consistent with the statistical plan published for the main trial [16] and detailed in Appendix B in S1 File. We fitted two models for intermediate outcomes. The minimally adjusted model included a random intercept for the cluster (village) and fixed effects for townships, baseline outcome, age, sex, and interview month. The fully adjusted model additionally accounted for variables significantly different ($P < 0.05$) between living patients and decedents (Table B in S1 File), and between surveyed and non-surveyed living patients (Table C in S1 File). All statistical analyses were performed with R version 4.1.1 (R Core Team [2021]; R Foundation for Statistical Computing, Vienna, Austria) and were separately replicated using Stata 17 (Stata Corporation, College Station, TX, USA). Statistical code was independently validated by biostatisticians at Duke University.

## Results

A total of 1,299 participants from 50 villages (mean age 65.7 years, 42.6% females) were recruited at baseline with 637 (from 25 villages) and 662 (from 25 villages) randomly allocated into the intervention arm and control arm. Among them 1,269 were alive at 12 months and eligible for post-trial follow-up (Fig 1). Most of participants (86.1%) had been diagnosed with ischemic stroke at baseline with a median duration of 5.3 years since the first stroke event. The baseline characteristic of participants in the intervention and control arms were generally comparable at baseline (Table 1).

Vital status was confirmed for all participants (Fig B in S1 File). During the initial 12-month trial phase, 30 deaths (2.3% of all participants) were recorded and the baseline characteristics of 1,269 surviving participants at 12 months was presented in Table 1. Additional 246 deaths (18.9% of all participants) were identified during post-trial period until 70-months post-baseline, resulting in a total of 276 deaths (21.2% of all participants, whose baseline characteristics was presented in Table D in S1 File). These resulted to a total of 6,890 person-years of follow-up (3,421 for intervention and 3,469 for control) for vital status. The median follow-up duration for the 276 deaths was 43.0 months (quantile 1–quantile 3: 26.7–56.8, range 2.3–70.0 months).

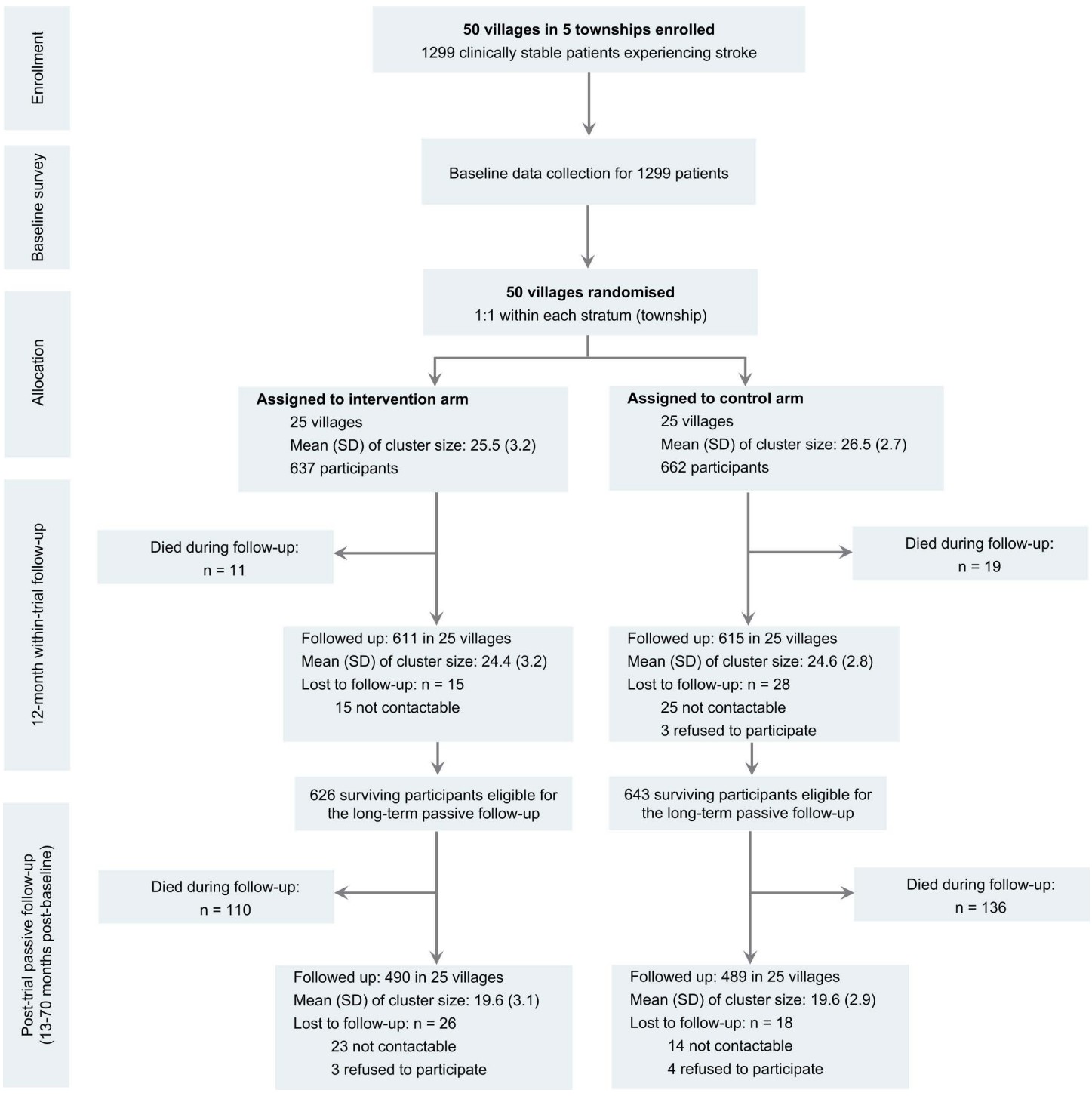

**Fig 1. Study profile of the SINEMA trial and the post-trial follow-up.** SD, standard deviation.

For the survived participants ($n = 1,023$, whose baseline characteristics presented in Table E in S1 File), duration of follow-up for vital status ranged from 69.0 to 70.2 months because all of them were surveyed during a 42-day period of recruitment and baseline survey in 2017 and followed-up until April 30, 2023. Notable differences of baseline characteristics were observed

**Table 1. Baseline characteristics of participants who recruited in the SINEMA trial ($n = 1,299$) and those who survived at 1-year post-trial for long-term follow-up participants ($n = 1,269$).**

| Characteristic | All participants* | | Survivors at 1-year for long-term follow-up[†] | | |
| --- | --- | --- | --- | --- | --- |
| | Intervention ($n = 637$) | Control ($n = 662$) | Intervention ($n = 626$) | Control ($n = 643$) | Overall ($n = 1,269$) |
| Age, years | 66.2 (8.2) | 65.2 (8.2) | 66.1 (8.2) | 65.0 (8.2) | 65.6 (8.2) |
| Sex | | | | | |
| Female | 272 (42.7%) | 281 (42.4%) | 268 (42.8%) | 278 (43.2%) | 546 (43.0%) |
| Male | 365 (57.3%) | 381 (57.6%) | 358 (57.2%) | 365 (56.8%) | 723 (57.0%) |
| Education | | | | | |
| No schooling | 264 (41.4%) | 274 (41.4%) | 261 (41.7%) | 266 (41.4%) | 527 (41.5%) |
| Primary school only | 182 (28.6%) | 205 (31.0%) | 177 (28.3%) | 198 (30.8%) | 375 (29.6%) |
| Above primary school | 191 (30.0%) | 183 (27.6%) | 188 (30.0%) | 179 (27.8%) | 367 (28.9%) |
| Marital status | | | | | |
| Married | 526 (82.6%) | 549 (82.9%) | 515 (82.3%) | 535 (83.2%) | 1,050 (82.7%) |
| Widowed, diveorced, or not married | 111 (17.4%) | 113 (17.1%) | 111 (17.7%) | 108 (16.8%) | 219 (17.3%) |
| Phone ownership | | | | | |
| No personal phone | 164 (25.7%) | 159 (24.0%) | 150 (23.3%) | 156 (24.9%) | 306 (24.1%) |
| Basic phone | 435 (68.3%) | 440 (66.5%) | 430 (66.9%) | 432 (69.0%) | 862 (67.9%) |
| Smart phone | 38 (6.0%) | 63 (9.5%) | 63 (9.8%) | 38 (6.1%) | 101 (8.0%) |
| Smoking status | | | | | |
| Current smoker | 99 (15.5%) | 122 (18.4%) | 98 (15.7%) | 119 (18.5%) | 217 (17.1%) |
| Former smoker | 130 (20.4%) | 132 (19.9%) | 127 (20.3%) | 123 (19.1%) | 250 (19.7%) |
| Never smoker | 408 (64.1%) | 408 (61.6%) | 401 (64.1%) | 401 (62.4%) | 802 (63.2%) |
| BMI, kg/m$^2$ | 25.5 (3.7) | 25.5 (3.6) | 25.4 (3.6) | 25.5 (3.7) | 25.5 (3.7) |
| Waist circumference, cm | 93.1 (10.0) | 93.1 (10.0) | 93.0 (9.9) | 93.1 (10.0) | 93.1 (10.0) |
| Stroke type | | | | | |
| Ischemic | 555 (87.1%) | 564 (85.2%) | 547 (87.4%) | 546 (84.9%) | 1,093 (86.1%) |
| Hemorrghage | 80 (12.6%) | 96 (14.5%) | 77 (12.3%) | 95 (14.8%) | 172 (13.6%) |
| Not specified | 2 (0.3%) | 2 (0.3%) | 2 (0.3%) | 2 (0.3%) | 4 (0.3%) |
| Stroke duration, years | | | | | |
| Since the first event | 5.2 (2.2–9.8) | 5.3 (2.4–9.8) | 5.1 (2.2–9.8) | 5.3 (2.4–9.9) | 5.2 (2.2–9.8) |
| Since the last event | 3.5 (1.1–6.8) | 3.2 (1.2–6.8) | 3.4 (1.1–6.7) | 3.2 (1.2–6.8) | 3.3 (1.1–6.8) |
| Self-report diseases | | | | | |
| Hypertension | 461 (72.4%) | 436 (65.9%) | 453 (72.4%) | 427 (66.4%) | 880 (69.3%) |
| Dyslipidemia | 248 (38.9%) | 271 (40.9%) | 245 (39.1%) | 266 (41.4%) | 511 (40.3%) |
| Diabetes | 113 (17.7%) | 103 (15.6%) | 112 (17.9%) | 101 (15.7%) | 213 (16.8%) |
| Heart diseases | 70 (11.0%) | 54 (8.2%) | 52 (8.1%) | 95 (9.3%) | 68 (10.9%) |
| Medication use | | | | | |
| Antiplatelet | 432 (67.8%) | 420 (63.4%) | 424 (67.7%) | 404 (62.8%) | 828 (65.2%) |
| Statin | 158 (24.8%) | 182 (27.5%) | 154 (24.6%) | 181 (28.1%) | 335 (26.4%) |
| Antihypertensive medicines | 522 (81.9%) | 508 (76.7%) | 511 (81.6%) | 495 (77.0%) | 1,006 (79.3%) |
| Adherence to medications[‡] | | | | | |
| Antiplatelet | 275/432 (63.7%) | 262/420 (62.4%) | 269/424 (63.4%) | 253/404 (62.6%) | 522/828 (63.0%) |
| Statin | 106/158 (67.1%) | 110/182 (60.4%) | 105/154 (68.2%) | 109/181 (60.2%) | 214/335 (63.9%) |
| Antihypertensive medicines | 329/522 (63.0%) | 316/508 (62.2%) | 322/511 (63.0%) | 308/495 (62.2%) | 630/1,006 (62.6%) |
| Physical activity, metabolic equivalents minutes per week | 1128.8 (346.5–2325.0) | 924.0 (240.0–2304.0) | 2037.0 (378.0–2664.0) | 2052.0 (297.5–2772.0) | 2044.5 (346.5–2706.0) |
| Systolic blood pressure, mmHg | 146.0 (20.9) | 145.7 (23.7) | 146.2 (21.0) | 145.9 (23.5) | 146.0 (22.3) |
| Diastolic blood pressure, mmHg | 78.0 (11.6) | 79.7 (11.7) | 78.1 (11.6) | 80.0 (11.6) | 78.9 (11.7) |

*(Continued)*

**Table 1.** (Continued)

| Characteristic | All participants* | | Survivors at 1-year for long-term follow-up† | | |
| --- | --- | --- | --- | --- | --- |
| | Intervention (*n* = 637) | Control (*n* = 662) | Intervention (*n* = 626) | Control (*n* = 643) | Overall (*n* = 1,269) |
| Timed up and go, completion time ≥14 s§ | 324 (51.6%) | 347 (53.1%) | 316 (50.5%) | 336 (52.3%) | 652 (51.4%) |
| Moderate to severe disability¶ | 179 (28.1%) | 173 (26.1%) | 172 (27.5%) | 166 (25.8%) | 338 (26.6%) |
| Stroke hospitalization in the past 12 months | 124 (19.5%) | 132 (19.9%) | 122 (19.5%) | 127 (19.8%) | 249 (19.6%) |

*Baseline characteristics stratified by intervention arm status replicate the results of our published paper [4].

†The 1-year intervention refers to the baseline characteristics of all surviving participants at the end of the 1-year intervention. Regardless of whether the participants completed the follow-up survey, they were included in the observational long-term follow-up.

‡Medication adherence was only measured among participants who were taking the specific medicine and was based on 4-item Morisky Green Levine Scale.

§Timed up and go results were recorded in seconds and dichotomized into binary as ≥14 s (indicating poorer limb mobility) versus <14 s (better limb mobility).

¶Disability was measured by the modified Rankin Scale, and patients who received a score above three were grouped into the "moderate-to-severe disability" group.

Data are *n* (%), *n/N* (%), mean (standard deviation), or median (Quantile1–Quantile3).

between the survivors and the deceased. Specifically, those who died were older on average, predominantly male, engaged in less physical activity at baseline, had poorer mobility, and exhibited a higher level of disability (Table F in S1 File).

Among the 276 deaths, 121 occurred in the intervention arm compared to 155 in the control arm (Table 2). About three in four deaths (75.7%) (*n* = 209; 92 in intervention *versus* 117 in control) had the underlying cause of cardiovascular diseases, and 32.6% of all deaths (43.1% of the cardiovascular cause-specific deaths) were due to stroke (*n* = 90; 38 in intervention *versus* 52 in control). As shown in Fig 2, between-arm differences in cumulative incidence favoring the intervention arm began to emerge around 12-months post-baseline (after the active intervention phase ended, and were previously reported) [4] and became more pronounced as time progressed for all-cause (Fig 2A), cardiovascular cause-specific (Fig 2B), and stroke cause-specific mortality (Fig 2C).

By 70-months post-baseline, the cumulative incidence of all-cause mortality reached 19.0% in the intervention arm versus 23.4% in the control arm, associated with a HR estimated from the Cox model of 0.73 (95% CI [0.59, 0.90]; *P* = 0.004; Table 2). Reduced risk in cardiovascular cause-specific mortality (HR, 0.73; 95% CI [0.58, 0.94]; *P* = 0.013) was also observed in the intervention arm, compared with control arm. Although lower cumulative incidence was observed in the intervention arm (6.0%) compared with the control arm(7.9%), for stroke cause-specific mortality, such differences was not statistically significant(HR was 0.71 (95% CI [0.44, 1.15]; *P* = 0.16)). The results remained robust after adjusting for variables identified as different between intervention arms at baseline (Table 2). In the subdistribution model, the results for the three mortality outcomes were consistent with the cause-specific model that treated competing events as censoring (Table G in S1 File).

The subgroup analysis on all-cause mortality showed that greater between-arm differences were observed with stronger association among participants who were female, younger than 65-years old, less educated, had lower household incomes, were divorced or widowed, had shorter stroke duration, had systolic blood pressure lower than 140 mmHg or had more severe disabilities at baseline (Fig 3). Significant differences in subgroups were observed for marital status (*P* for interaction = 0.002), education level, and disability status (both *P* for interactions < 0.10). The observed patterns for subgroup analyses of both cardiovascular (Fig C in S1 File) and stroke cause-specific mortality (Fig D in S1 File) mirrored those seen for all-cause mortality.

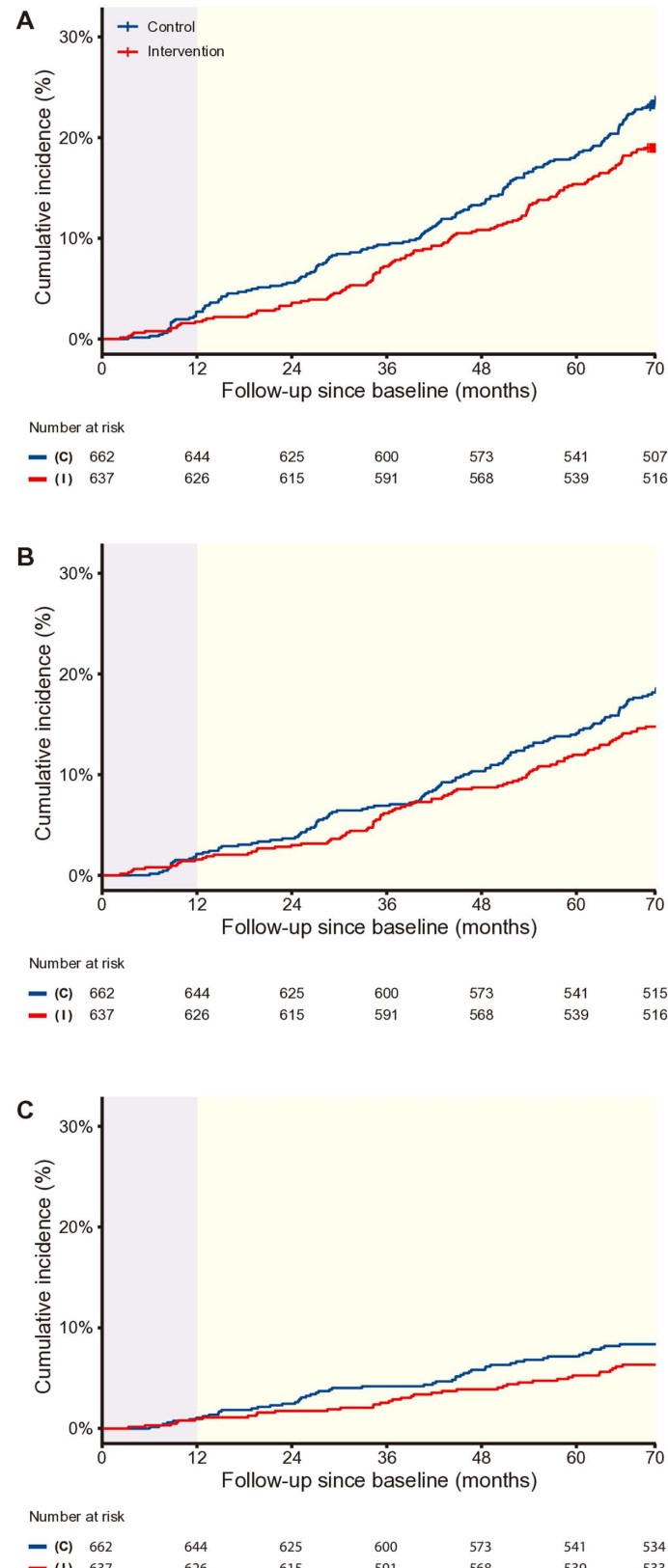

**Fig 2. Cumulative incidence for (A) all-cause, (B) cardiovascular cause-specific, and (C) stroke cause-specific mortality from baseline to 70 months.** Light purple area indicates 12-month active intervention phase. Light yellow

area indicates post-trial observational phase. (C) indicates control arm receiving usual care (blue lines) and (I) indicates arm receiving the SINEMA intervention, a primary care-based mobile health intervention for stroke secondary prevention (orange lines).

**Table 2. The cumulative incidence of mortality and hazard ratios (95% CIs) of intervention on mortality at 70 months post-baseline ($n = 1,299$).**

| | Intervention ($n = 637$) | | Control ($n = 662$) | | Minimally adjusted HR (95% CI)* | P-value | Fully adjusted HR (95% CI)† | P-value |
|---|---|---|---|---|---|---|---|---|
| **Outcomes** | *N* (cumulative incidence, %) | Death rates (per 1,000 person-year) | *N* (cumulative incidence, %) | Death rates (per 1,000 person-year) | | | | |
| **All-cause mortality** | 121 (19.0%) | 35.4 | 155 (23.4%) | 44.7 | 0.73 (0.59, 0.90) | 0.004 | 0.72 (0.58, 0.90) | 0.004 |
| **Cardiovascular cause-specific mortality‡** | 92 (14.4%) | 26.9 | 117 (17.7%) | 33.7 | 0.73 (0.58, 0.94) | 0.013 | 0.73 (0.56, 0.93) | 0.012 |
| Stroke | 38 (6.0%) | 11.1 | 52 (7.9%) | 15.0 | 0.71 (0.44, 1.15) | 0.16 | 0.75 (0.46, 1.21) | 0.24 |
| Non-stroke | 54 (8.5%) | 15.8 | 65 (9.8%) | 18.7 | 0.75 (0.53, 1.07) | 0.12 | 0.71 (0.51, 1.01) | 0.06 |
| **Other causes of mortality§** | 29 (4.6%) | 8.5 | 38 (5.7%) | 11.0 | 0.71 (0.47, 1.08) | 0.11 | 0.72 (0.46, 1.11) | 0.14 |

*HRs were adjusted for town, age, and sex.

†HRs were adjusted for town, age, sex, and variables noted to be differential by intervention arms at baseline (baseline diastolic blood pressure, having hypertension, having none of the assessed home assets [TV, refrigerator, air conditioner, and computer], and taking antihypertensive medicines).

‡The classification of cardiovascular (including stroke and non-stroke) cause-specific mortality is based on the Global Burden of Diseases, Injuries, and Risk Factors Study [12].

§The specific other causes of mortality are detailed in Table A is S1 File.

N, number of deaths; HR, hazard ratio; CI, confidence interval.

To better understand mortality outcomes, we evaluated the intermediate outcomes among participants who survived at long-term follow-up, assessed in 2022−23 (median follow-up of 66.6 months post baseline). Out of the 1,023 confirmed survivors, loss to follow up was low ($n = 44$, 4.3%) and 979 successfully completed the survey and were included in the intermediate outcome analysis. Over 66 months, the SINEMA intervention was associated with a sustained and significant net reduction in systolic blood pressure of 2.9 mmHg (95% CI, −5.4, −0.4; Table 3). Observed differences favoring intervention was observed also for diastolic blood pressure and medication adherence with higher proportion of individuals with perfect adherence to medication, but not for physical activity or functioning abilities (Table 3; risk differences are reported in Table H in S1 File). Post-hoc analysis on changes in potential lifestyle factors, including smoking status, body mass index (BMI), and waist circumference were reported in Table I in S1 File, but no significant differences in two arms were observed.

## Discussion

During a maximum follow-up of 70.2 months from the baseline of the SINEMA trial, individuals who received the SINEMA intervention showed significant long-term relative reductions of 27% for all-cause mortality, 27% for cardiovascular cause-specific mortality. Although lower incidence in stroke cause-specific mortality was observed in the intervention arm, the between-arm difference was not statistically significant. These between-arm differences were descriptively more pronounced among females, less educated, divorced or widowed, disabled, or lower-income people.

Our study is one of the few long-term observational follow-up studies of cRCTs on patients experiencing stroke in LMICs. The 70-month follow-up analyses extend earlier findings on the effectiveness of primary care-based mHealth strategies to promote stroke secondary

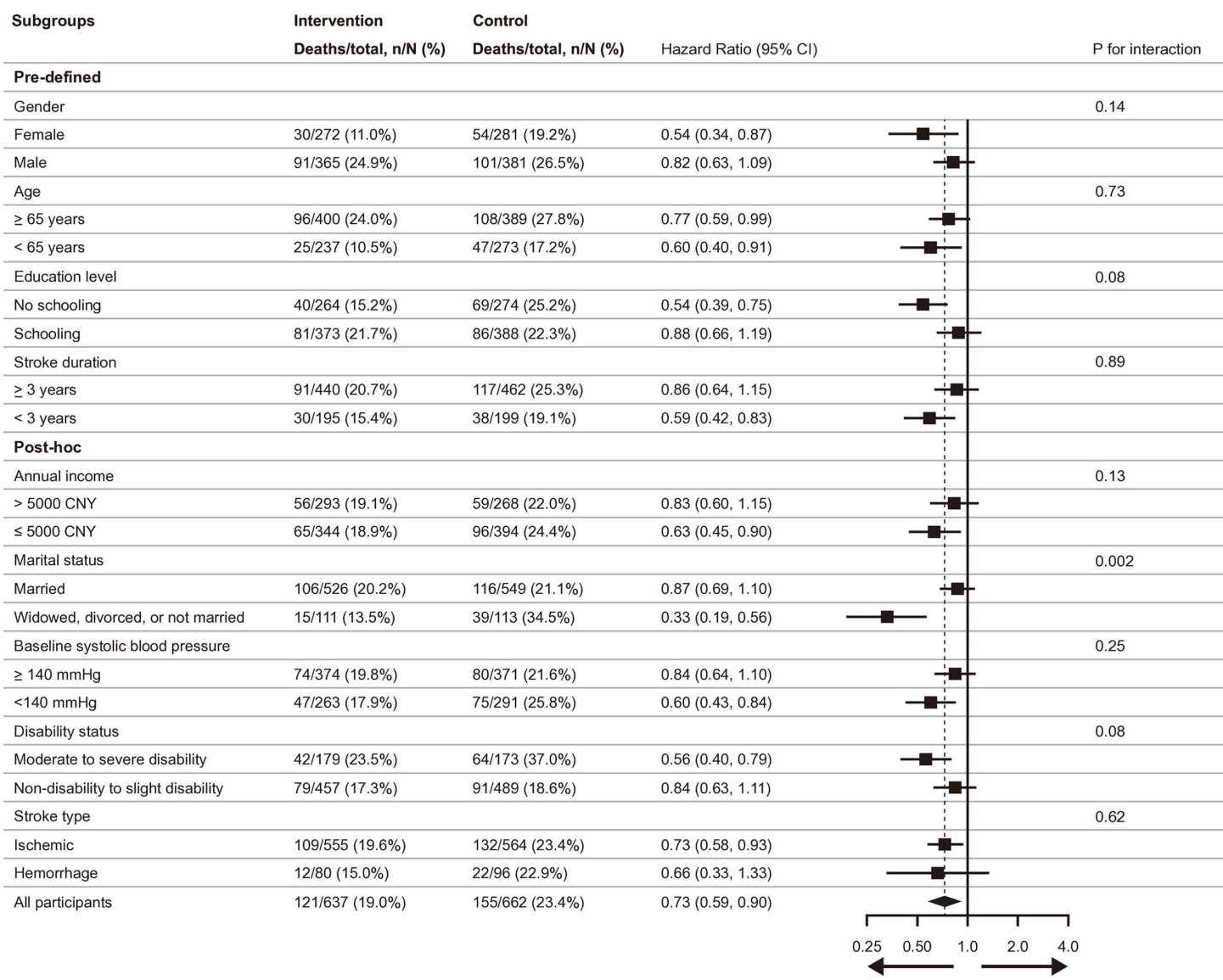

| Subgroups | Intervention | Control | | | |
|---|---|---|---|---|---|
| | Deaths/total, n/N (%) | Deaths/total, n/N (%) | Hazard Ratio (95% CI) | | P for interaction |
| **Pre-defined** | | | | | |
| Gender | | | | | 0.14 |
| Female | 30/272 (11.0%) | 54/281 (19.2%) | 0.54 (0.34, 0.87) | | |
| Male | 91/365 (24.9%) | 101/381 (26.5%) | 0.82 (0.63, 1.09) | | |
| Age | | | | | 0.73 |
| ≥ 65 years | 96/400 (24.0%) | 108/389 (27.8%) | 0.77 (0.59, 0.99) | | |
| < 65 years | 25/237 (10.5%) | 47/273 (17.2%) | 0.60 (0.40, 0.91) | | |
| Education level | | | | | 0.08 |
| No schooling | 40/264 (15.2%) | 69/274 (25.2%) | 0.54 (0.39, 0.75) | | |
| Schooling | 81/373 (21.7%) | 86/388 (22.3%) | 0.88 (0.66, 1.19) | | |
| Stroke duration | | | | | 0.89 |
| ≥ 3 years | 91/440 (20.7%) | 117/462 (25.3%) | 0.86 (0.64, 1.15) | | |
| < 3 years | 30/195 (15.4%) | 38/199 (19.1%) | 0.59 (0.42, 0.83) | | |
| **Post-hoc** | | | | | |
| Annual income | | | | | 0.13 |
| > 5000 CNY | 56/293 (19.1%) | 59/268 (22.0%) | 0.83 (0.60, 1.15) | | |
| ≤ 5000 CNY | 65/344 (18.9%) | 96/394 (24.4%) | 0.63 (0.45, 0.90) | | |
| Marital status | | | | | 0.002 |
| Married | 106/526 (20.2%) | 116/549 (21.1%) | 0.87 (0.69, 1.10) | | |
| Widowed, divorced, or not married | 15/111 (13.5%) | 39/113 (34.5%) | 0.33 (0.19, 0.56) | | |
| Baseline systolic blood pressure | | | | | 0.25 |
| ≥ 140 mmHg | 74/374 (19.8%) | 80/371 (21.6%) | 0.84 (0.64, 1.10) | | |
| <140 mmHg | 47/263 (17.9%) | 75/291 (25.8%) | 0.60 (0.43, 0.84) | | |
| Disability status | | | | | 0.08 |
| Moderate to severe disability | 42/179 (23.5%) | 64/173 (37.0%) | 0.56 (0.40, 0.79) | | |
| Non-disability to slight disability | 79/457 (17.3%) | 91/489 (18.6%) | 0.84 (0.63, 1.11) | | |
| Stroke type | | | | | 0.62 |
| Ischemic | 109/555 (19.6%) | 132/564 (23.4%) | 0.73 (0.58, 0.93) | | |
| Hemorrhage | 12/80 (15.0%) | 22/96 (22.9%) | 0.66 (0.33, 1.33) | | |
| All participants | 121/637 (19.0%) | 155/662 (23.4%) | 0.73 (0.59, 0.90) | | |

0.25   0.50   1.0   2.0   4.0

Favour intervention  Favour control

**Fig 3. All-cause mortality in each pre-specified and post-hoc subgroup, minimally adjusted model.** Hazard ratios were adjusted for age, sex, and township, with villages considered as clusters. CI, confidence interval.

prevention in resource-limited settings [2–4] and further proven the potential sustained impact of this time-limited intensive intervention on mortality.

Stroke was responsible for 6.55 million excess deaths in 2019 worldwide[1], with direct healthcare expenditures reaching US$183.13 billion [18]. We also found a cumulative incidence of overall mortality as high as 21.2% at 5.5 years, comparable to the 21% death rate at 6 years in a large Chinese population study [19]. The long-term outcomes in mortality observed in this study address such global health challenges and shed light for effective strategies to reduce stroke mortality especially in resource-poor settings.

This long-term follow-up of SINEMA trial addresses the gap in assessing the sustained association of interventions with mortality. Previous trials typically reported within-trial findings or focused on the immediate outcomes of cardiovascular risk factors such as blood

**Table 3. Effects of the SINEMA intervention on intermediate outcomes from baseline to 70 months among living patients ($n = 979$).**

| Outcomes | Intervention arms | | Minimally adjusted model** | Fully adjusted model*** |
|---|---|---|---|---|
| | Intervention* ($n = 490$) | Control ($n = 489$)* | Estimate (95% CI)† | Estimate (95% CI)† |
| **Blood pressure (mmHg)** | | | | |
| Change in systolic blood pressure | −4.3 (23.4) | −2.2 (24.2) | −2.9 (−5.4, −0.4) | −3.0 (−5.7, −0.4) |
| Change in diastolic blood pressure | 3.1 (11.3) | 3.4 (11.1) | −1.3 (−2.6, 0.0) | −1.2 (−2.5, 0.1) |
| **Medication adherence‡** | | | | |
| Antiplatelets | 64.9% (198) | 59.4% (174) | 1.10 (0.95, 1.26) | 1.09 (0.94, 1.26) |
| Statins | 63.9% (131) | 61.0% (136) | 1.07 (0.89, 1.28) | 1.08 (0.92, 1.28) |
| Antihypertensive medicines | 67.0% (286) | 61.0% (246) | 1.10 (0.99, 1.22) | 1.08 (0.98, 1.20) |
| **Physical activity and functionning** | | | | |
| Change in physical activity, metabolic equivalents minutes per week | −541.3 (1876.8) | −365.7 (1855.1) | −94.2 (−313.2, 124.8) | −46.9 (−271.1, 177.2) |
| Timed up and go ≥14 s§ | 62.7% (279) | 59.9% (266) | 1.04 (0.93, 1.16) | 1.02 (0.91, 1.13) |

Data are % ($n$), or mean (standard deviation). CI, confidence interval.

*At 70-months post-baseline, there were a total of 1,023 living patients, of whom 979 completed the long-term follow-up survey. The remaining 44 participants were either unreachable or declined to participate. The intermediate outcome analysis included only the 979 individuals who completed the survey.

**Minimally adjusted model: Adjusted for baseline outcome, township, sex, age, and interview month; removing outliers (2 for systolic blood pressure, 6 for diastolic blood pressure) in the outcome variable (based on a priori decision to remove those that are more than 2 interquartile ranges above the third quartile or below the first quartile).

***Fully adjusted model: Adjusted for baseline outcome, township, sex, age, interview month, variables noted to be differential by treatment arm at baseline ($p < 0.05$; baseline diastolic blood pressure, having hypertension, having none of home assets [TV, refrigerator, air conditioner, and computer], and taking antihypertensive medicines), and baseline variables related to death (health-related quality of life, whether achieve health enhancing physical activity, stroke recurrence, disability, depression status, and completion time of timed up and go test) and loss-to-follow-up among participants (phone ownership, smoking statu, and stroke duration); removing the same outliers as in the minimally adjusted model.

†For continuous outcomes (systolic blood pressure, diastolic blood pressure, and physical activity), "estimate" refers to the differences between the arms in 70-month change since baseline (control arm is the reference); for binary outcomes (timed up and go, medication adherence), "estimate" refers to the risk ratio (control arm is the reference).

‡Medication adherence refers to the percentage of patients with a perfect adherence which refers to a score of 0 on the 4-item Morisky Green Levine Scale. Medication adherence was only measured among participants who were taking antiplatelets (intervention $n = 305$, control = 293), statins ($n = 205$, $n = 223$), and antihypertensive medicines ($n = 427$, $n = 403$). Medication adherence outcomes were not adjusted for baseline outcome, since the set of participants taking a given medication at baseline was not the same set taking the medicines at the post-trial follow-up.

§"Timed up and go test" results were recorded in seconds during measurement and dichotomized into binary as ≥14 s (indicating poorer limb mobility) versus <14 s (better limb mobility).

pressure [20,21] or cardiovascular events [22]. While a few trials on hypertension control, such as COBIN [23] and COBRA [24] reported 5-year post-trial effects on the outcome of blood pressure, they yielded mixed findings [23]. In addition, the long-term effects of interventions involving mHealth technology on cardiovascular disease management remain unreported. Our long-term follow-up study observed sustained association of the intervention with some intermediate outcomes [13], as well as on long-term mortality. The significant relative reductions of over 20% in deaths provide evidence for policy-makers to scale up the SINEMA intervention to reap larger population health benefits. Of note, the delivery cost was relatively low at US$24 per capita [4], with high cost-effectiveness during the trial (incremental cost-effectiveness ratio of US$792.9 per quality-adjusted life year gained) and cost savings post-trial (protocol published; results under review) [25], promising for transfer and broader implementation. Discussions are ongoing to engage policy-makers and other stakeholders in potential scale-up.

The relative reduction in mortality observed in our study may seem larger than what can be expected from the reduction in blood pressure. Although the net reduction of 2.9 mmHg in systolic blood pressure appeared modest, this reduction was sustained over the long term of 6

years. Although multiple factors and mechanisms could contribute to the changes in mortality and our complex intervention is not fully designed to illustrate such mechanisms of changes, this sustained association was likely associated with the maintained behavior changes in medication adherence in the intervention arm, which may provide some clues. A meta-analysis of 18 clinical trials demonstrated that interventions enhancing medication adherence reduce mortality in patients with cardiovascular diseases [26]. Evidence also indicated that even a 1 mmHg reduction in systolic blood pressure significantly decreases the risk of fatal and non-fatal cardiovascular events and stroke, thereby reducing mortality [26].

The substantial reduction in mortality may be partially attributable to the higher death risk in older rural Chinese patients experiencing stroke. Despite improvements in healthcare, stroke mortality in China still remains higher than in many high-income countries [27,28]. In addition, urban-rural disparities in stroke mortality have persisted—14% *versus* 21% in 2020 [19]. Therefore, our study has the potential to reduce urban–rural inequity in health outcomes for patients experiencing stroke. Beyond this, the SINEMA intervention also shows promise in reducing health inequity by socioeconomic status or sex. Our study observed stronger between arm differences than the overall effect size among vulnerable groups with poor access to optimal care, including females, the less educated, divorced or widowed individuals, the disabled, and those with lower income [29]. These results highlight the greater value of investment in primary care to reduce health inequalities linked to residence, socioeconomic status, or sex [30].

Interestingly, we also observed reduction in non-stroke cardiovascular cause-specific mortality, indicating the potential broader cardiovascular benefits beyond stroke prevention. Such observed reduction in non-stroke cardiovascular cause-specific mortality may be attributable to the control of shared risk factors between stroke and other cardiovascular diseases in blood pressure, improvement in medication adherence to statins and antiplatelet use and increase in physical activity, though more studies are needed to investigate such broader benefits.

Several key features of the SINEMA intervention may have contributed to its long-term sustained association: (1) the design of the intervention—based on comprehensive field-based contextual research [5,7,8] and streamlined clinical guidelines suitable for resource-limited settings—increased the likelihood of sustained effectiveness; (2) delivered by village doctors, the intervention integrated into routine care with low marginal cost of human resources. Moreover, motivations from non-financial incentives, as well as the trust building with patients as reported in the process evaluation, may have lasting association beyond the trial period [31]; (3) the intervention was human-centerd with equal emphases on behavior changes for both patients and providers instead of only one group. Monthly follow-up visits reinforced provider–patient interactions—a core principle of the chronic care model [32]; and (4) mHealth technology was crucial to our intervention [33]. The SINEMA app, free for village doctors, featured user-friendly functions could be used after the trial stopped.

Key strengths of our study include extensive field-based contextual research and rigorously implemented cRCT design, extended post-trial follow-up, intention-to-treat analyses with independent biostatistical verification, and standardized endpoint assessment from multiple sources to ensure objectivity and complete (100%) follow-up on vital status. Our study also has several limitations. First, the study was conducted in a single county in Northern China, which limits its generalizability. Future research using a qualitative approach is being planned to provide insight on adaptable strategies for other settings. Second, although we have tried to infer the roles of several factors through intermediate outcome analyses, we were unable to fully uncover the exact mechanisms by which the multicomponent intervention reduced mortality. Moreover, given the observational nature of the post-trial follow-up phase and the potential influence of unobserved interventions or unmeasured factors on mortality,

we are not able to fully build the causal inferences, and caution should be exercised when interpreting the findings. Lastly, not all causes of death have accurate clinical diagnostic records in rural areas of China, but our multi-source verification partially mitigates this concern. More importantly, the assessment methods remained identical for both the intervention and the control arms, ensuring their compatibility.

In conclusion, this study demonstrated a sustained reduction in long-term mortality of the primary care-based mHealth intervention beyond the trial phase among patients experiencing stroke in rural China. These results provide clinically and practically implication for evidence-based policymaking, suggesting the potential of improving stroke secondary prevention in resource-limited areas and reducing health inequity via strengthening primary care and applying digital health solutions. Such intervention strategies are potentially relevant to not only stroke but other non-communicable diseases. To aid its expansion and wider implementation, future research investigating the transferability and adaptation of the implementation strategies in China and other LMICs are warranted. When scaled up, the intervention has the potential to contribute to achieving the sustainable development goals of reducing premature non-communicable disease mortality by 2030.

## Supporting information

**S1 Protocol. Study protocol as approved by the ethical committee in China and USA.** (DOCX)

**S1 File. Fig A.** Timeline for trial phase and post-trial follow-up phase in SINEMA. **Table A.** Full list of mortality outcomes with ICD-10 codes used in the present study. **Appendix A.** Definitions, measurement for intermediate outcomes in S1 File. **Appendix B.** Statistical analysis plan for intermediate outcomes in S1 File. **Table B.** Baseline characteristics of participants who were followed and deceased at 70-month follow-up ($n = 1,255$). **Table C.** Baseline characteristics by status of follow-up among participants who survived at 70 months ($n = 1,023$). **Fig B.** Flow diagram for confirming mortality information from three sources. **Table D.** Baseline characteristics of participants of the SINEMA trial who died during 70-months follow-up post-baseline ($n = 276$). **Table E.** Baseline characteristics by intervention arms of participants who survived at 70-months post-baseline ($n = 1,023$). **Table F.** Baseline characteristics of participants in the SINEMA trial by vital status at 70-months post-baseline ($n = 1,299$). **Table G.** The adjusted hazard ratios (95% confidence intervals) of intervention on mortality in patients experiencing stroke at 70-month follow-up using sub-distribution model. **Fig C.** Long-term outcomes on cardiovascular cause-specific mortality in each pre-specified and post-hoc subgroup, minimally adjusted model. **Fig D.** Long-term outcomes on stroke cause-specific mortality in each pre-specified and post-hoc subgroup, minimally adjusted model. **Table H.** Effects (risk difference) of the SINEMA intervention on intermediate outcomes from baseline to 70 months among living patients ($n = 979$). **Table I.** Participants characteristics and intermediate outcomes at baseline and at the long-term follow-up ($n = 979$). (DOCX)

**S1 Checklist. CONSORT 2010 checklist for reporting a cluster randomised trial.** (DOCX)

## Acknowledgments

We thank Zhenli Xu, Mobai Hou, and the team at the Nanhe District Centre for Disease Prevention and Control, Li Wang, Huayun Zhang and the team at the Renze District Centre for Disease Prevention and Control, and all staff members from township healthcare centres and

village clinics, and all patients who participated or supported the study. We would like to also acknowledge Jun Wang (Department of Otorhinolaryngology, Head, and Neck Surgery, The First Affiliated Hospital of Nanchang University)'s help for data visualization.

## Author contributions

**Conceptualization:** Enying Gong, Lijing L. Yan.

**Data curation:** Xingxing Chen, Jie Tan, John A. Gallis, Shifeng Sun, Siran Luo, Bolu Yang, Yutong Long.

**Formal analysis:** Xingxing Chen, Jie Tan, Elizabeth L. Turner, John A. Gallis.

**Funding acquisition:** Enying Gong, Ruitai Shao, Lijing L. Yan.

**Investigation:** Siran Luo, Fei Wu, Bolu Yang.

**Methodology:** Enying Gong, Elizabeth L. Turner, John A. Gallis, Shifeng Sun, Janet P. Bettger, Brian Oldenburg.

**Project administration:** Xingxing Chen, Siran Luo, Fei Wu.

**Resources:** Yilong Wang, Yun Zhou.

**Software:** Elizabeth L. Turner, John A. Gallis, Fei Wu.

**Supervision:** Yilong Wang, Zixiao Li, Yun Zhou, Shenglan Tang, Janet P. Bettger, Brian S. Mittman, Valery L. Feigin, Ruitai Shao, Lijing L. Yan.

**Validation:** Jie Tan, Elizabeth L. Turner, Shifeng Sun.

**Writing – original draft:** Xingxing Chen.

**Writing – review & editing:** Enying Gong, Brian Oldenburg, Xiaochen Zhang, Jianfeng Gao, Brian S. Mittman, Valery L. Feigin, Shah Ebrahim, Lijing L. Yan.

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
