## [Editor Report · Decision Letter 0]

23 Oct 2024

Dear Dr Yan,

Thank you for submitting your manuscript entitled "Long-term effect on mortality of a primary care-based mobile health intervention for stroke: six-year follow-up of a cluster-randomized controlled trial" for consideration by PLOS Medicine.

Your manuscript has now been evaluated by the PLOS Medicine editorial staff and I am writing to let you know that we would like to send your submission out for external peer review.

During the assessment of your manuscript, confusion arose regarding the relevant protocol/registration for the submitted study. Could you please comment and clarify whether

1) the current study is part of the SINEMA study (e.g. the exploratory outcomes), and if so, why the outcomes are not 100% identical, or

2) the study is part of NCT05792618 (SaFaRI study), or

3) it is a post-hoc analysis according to the study protocol provided in Supplement 2.docx.

Please feel free to email me at atosun@plos.org with your response or include a response when you resubmit your manuscript.

Please re-submit your manuscript within two working days, i.e. by Oct 25 2024.

Feel free to email me at atosun@plos.org or us at plosmedicine@plos.org if you have any queries relating to your submission.

Kind regards,

Alexandra Tosun, PhD

Associate Editor

PLOS Medicine

---

## [Decision Letter · Decision Letter 1]

16 Dec 2024

Dear Dr Yan,

Many thanks for submitting your manuscript "Long-term effect on mortality of a primary care-based mobile health intervention for stroke: six-year follow-up of a cluster-randomized controlled trial" (PMEDICINE-D-24-03550R1) to PLOS Medicine. The paper has been reviewed by subject experts and a statistician; their comments are included below and can also be accessed here: [LINK]

As you will see, the reviewers found the study to be well reported, but raised several points for clarification. After discussing the paper with the editorial team and an academic editor with relevant expertise, I'm pleased to invite you to revise the paper in response to the reviewers' comments. We plan to send the revised paper to some or all of the original reviewers, and we cannot provide any guarantees at this stage regarding publication.

We ask that you submit your revision by Jan 06 2025. However, if this deadline is not feasible (including due to the upcoming holidays), please contact me by email and we can discuss a suitable alternative. Please note that the editorial team will be out of office from 23 December 2024 up to and including 3 January 2025.

Don't hesitate to contact me directly with any questions (atosun@plos.org).

Best regards,

Alexandra

Alexandra Tosun, PhD

Associate Editor

PLOS Medicine

atosun@plos.org

Comments from the editorial team:

1) In your response regarding study affiliation, you stated that the current study is an extension of the SINEMA study, but also that the current manuscript is part of the SaFaRI study. Please clarify this and ensure that it is clearly described in the Methods section.

2) In the study protocol you stated that the secondary outcomes of interest will be in line with the original protocol. We are aware that some of the long-term outcomes have already been reported in a separate publication. We ask you to transparently list all outcomes in the Methods section and describe which outcomes are reported in the current submission, have been or will be reported elsewhere. In line with this, it will be important to clearly describe which outcomes were pre-specified and which were added post hoc. We would also encourage you to include all remaining outcomes, such as health-related quality of life.

Comments from the reviewers:

Reviewer #1: Statistical review

This paper reports a long-term follow-up of a cluster randomised trial which assessed a primary-care intervention. The study shows the intervention had long-term beneficial effects on a number of important outcomes. Overall the methods used are appropriate and the results reported well. I have a few comments which I have listed below:

1. Abstract: the abstract mentions a number of subgroup analyses. Some of these were posthoc according to the methods, so I would generally recommend not to mention them in the abstract (or at least label them as so). It is more appropriate to include the pre-specified subgroups in the abstract but it seems to me that none of these had significant interactions: I'd therefore not recommend highlighting any of the subgroup analyses in the abstract unless I got any of the above wrong.

2. Methods: the registration page for the long-term follow-up (NCT05792618) mentions a lot of outcomes that aren't reported here - it might be good to clarify that this paper is not reporting those outcomes.

3. Methods/results: I did not follow how the number of survivors was so similar in each arm in the intermediate outcome analysis given that there was a difference in mortality between groups - can this be explained?

James Wason

Reviewer #2: This is a study reporting post-intervention long-term outcomes of a 12-month intervention namely system-integrated technology-enabled model of care for stroke survivors. Up to 70 months of follow-up revealed the SINEMA intervention reduced all-cause mortality, and cardiovascular mortality than the control arm. This is a well written paper with important findings that are particularly relevant to low income regions of the globe.

Comments

None

Reviewer #3: In the manuscript "Long-term effect on mortality of a primary care-based mobile health intervention for stroke: six-year follow-up of a cluster-randomized controlled trial" the authors have evaluated the long-term (70 months) effect of a primary care-based integrated mobile health intervention for secondary prevention of stroke on all-cause and cardiovascular mortality in rural China. The original intervention was tested in a cluster randomised controlled trial (SINEMA) which evaluated the effect of the intervention on blood pressure reduction.

While the study provides evidence for long term effect of this intervention on mortality, I have several concerns.

Major comments

1. One major concern is the way the data are presented. A key question is what has led to differences in mortality in the two arms after the follow up period. The answer to this question is not very clear in the data presented. One would like to know how the baseline characteristics varied in the follow up cohort in the two arms. The authors should provide the details of these characteristics for surviving individuals in the intervention and control arms in table. The data for the decedents should also be presented for both the arms separately.

2. In addition to smoking tobacco, the authors should present data on other risk factors for stroke such as alcohol use, salt use or awareness about the harms of high salt use and BMI in both the arms during follow up if they have these data.

3. The authors should provide a flow diagram of the number of patients enrolled, survived, died and migrated in both the arms during the initial trial period and subsequent to this period.

4. The authors refer to the original paper for the details of the intervention but in the interest of the readers it will be important to describe these in brief. For example, the components of mobile health intervention and lifestyle interventions should be described in a few sentences in this manuscript as well.

5. Table 2 shows that in the cardiovascular mortality, it is the "non-stroke mortality" that is lower in the intervention arm. The authors might want to discuss potential reason for this as the intervention was primarily designed for stroke.

6. The authors have attributed the reduction is mortality primarily to medication adherence and modest reduction in blood pressure but have not considered effect of lifestyle changes. However, this is not very clear to me based on the data presented. This will only be true if the risk factor/lifestyle change distribution is not different in the two arms during the follow up period. By presenting the data as asked for in point 1 a reader may be able to infer better.

Minor comments

1. Table 3, the number of patients included in the analysis is 979 but the number of survivors is 1023. The authors may want to provide an explanation on the footnote for this difference.

2. In table 3, the explanation for the data shown as percentages for medication adherence is not very clear. The authors have provided a note "‡ Medication adherence refers to a perfect adherence with score of 0 based on the 4-item Morisky Green Levine Scale." The wording may be changed to-"‡ Medication adherence refers to the percentage of patients with a perfect adherence which refers to a score of 0 on the 4-item Morisky Green Levine Scale."

---

* Please upload any figures associated with your paper as individual TIF or EPS files with 300dpi resolution at resubmission; please read our figure guidelines for more information on our requirements: http://journals.plos.org/plosmedicine/s/figures. While revising your submission, please upload your figure files to the PACE digital diagnostic tool, https://pacev2.apexcovantage.com/. PACE helps ensure that figures meet PLOS requirements. To use PACE, you must first register as a user. Then, login and navigate to the UPLOAD tab, where you will find detailed instructions on how to use the tool. If you encounter any issues or have any questions when using PACE, please email us at PLOSMedicine@plos.org.

* ETHICS STATEMENTS: Please provide details of the consent and whether it was written or oral. Please also provide the approval numbers.

* COMPETING INTERESTS: All authors must declare their relevant competing interests per the PLOS policy, which can be seen here: https://journals.plos.org/plosmedicine/s/competing-interests

For authors with ties to industry, please indicate whether any of the interests has a financial stake in the results of the current study.

* DATA AVAILABILITY: PLOS Medicine requires that the de-identified data underlying the specific results in a published article be made available, without restrictions on access, in a public repository or as Supporting Information at the time of article publication, provided it is legal and ethical to do so. Please see the policy at http://journals.plos.org/plosmedicine/s/data-availability

and FAQs at http://journals.plos.org/plosmedicine/s/data-availability#loc-faqs-for-data-policy

The Data Availability Statement (DAS) requires revision. For each data source used in your study:

FIGURES AND TABLES

SUPPLEMENTARY MATERIAL

REFERENCES

STUDY TYPE-SPECIFIC REQUESTS

* Please be explicit in the title and the abstract that is a post-hoc sub-study/follow-on study of the primary trial. We suggest reporting in-line with CONSORT explicitly stating the sub-study nature and ensuring that the abstract details the main trial items in 2-3 sentences, including the study population, dates, intervention and primary outcome. The majority of the abstract should then describe the complete details of this post-hoc sub-study.

* Please complete the CONSORT checklist and ensure that all components of CONSORT are present in the manuscript as well as clearly defined details of this sub-study. When completing the checklist, please use section and paragraph numbers, rather than page numbers as these often change in the event of publication.

* Please ensure that study registration details are included in the Methods section.

* Abstract: Please include the study design, population and setting, number of participants, years during which the study took place (original enrollment and follow up), length of follow up, and main outcome measures.

* Please include absolute numbers wherever you report percentages; eg, n/N (%)

* In keeping with our commitment to Open Science, please include the study protocol document and analysis plan (including any amendments) as Supporting Information to be published with the manuscript if accepted.

---

## [Decision Letter · Decision Letter 2]

31 Jan 2025

Dear Dr. Yan,

Thank you very much for re-submitting your manuscript "Long-term effect on mortality of a primary care-based mobile health intervention for stroke: six-year follow-up of a cluster-randomized controlled trial" (PMEDICINE-D-24-03550R2) for review by PLOS Medicine.

Thank you for your detailed response to the editors' and reviewers' comments. I have discussed the paper with my colleagues, and it has also been seen again by two of the original reviewers. The changes made to the paper were mostly satisfactory to the reviewer. As such, we intend to accept the paper for publication, pending your attention to the reviewers' and editors' comments below in a further revision. When submitting your revised paper, please once again include a detailed point-by-point response to the editorial comments.

[LINK]

In revising the manuscript for further consideration here, please ensure you address the specific points made by each reviewer and the editors. In your rebuttal letter you should indicate your response to the reviewers' and editors' comments and the changes you have made in the manuscript. Please submit a clean version of the paper as the main article file. A version with changes marked must also be uploaded as a marked up manuscript file. Please also check the guidelines for revised papers at http://journals.plos.org/plosmedicine/s/revising-your-manuscript for any that apply to your paper.

We ask that you submit your revision within 1 week (Feb 07 2025). However, if this deadline is not feasible, please contact me by email, and we can discuss a suitable alternative.

Please do not hesitate to contact me directly with any questions (atosun@plos.org). If you reply directly to this message, please be sure to 'Reply All' so your message comes directly to my inbox.

We look forward to receiving the revised manuscript.

Sincerely,

Alexandra Tosun, PhD

Associate Editor 

PLOS Medicine

plosmedicine.org

Comments from Reviewers:

Reviewer #1: Thank you to the authors for addressing my previous comments well. I have no further issues to raise.

Reviewer #3: The authors have not provided the demographic features of the participants in the cohort for this follow up study. The cohort for this follow up study has 626 survivors in the intervention arm and 643 in the control arm as shown in Figure 1. I have raised this concern in the earlier review as well. This is very important as these individuals form the baseline for the follow up phase described in this manuscript. They should present this data in table 1. The data on survivors and decedents in table 1 can be moved to supplementary material. The authors have addressed my other concerns.

[LINK]

Requests from Editors:

GENERAL

Your study is observational and therefore causality cannot be inferred. Please remove any language that implies causality, such as effect. Please refer to associations instead. Please revise throughout the entire manuscript including figures and tables.

Given that the intervention was not continued after the 12-month intervention period, we ask you to temper the findings and conclusions of your study. Within the 58-month time frame following the active trial, other factors not accounted for may have influenced the associations observed in this study, i.e., the sustained link you observe may not be a result of the intervention. Accordingly, we ask you to expand the limitations section to consider factors such as the survey-based nature of your study and the confounders that may not have been accounted for.

The terms gender and sex are not interchangeable (as discussed in https://www.who.int/health-topics/gender#tab=tab_1 ); please use the appropriate term and revise accordingly throughout the manuscript. Please note that you currently use both.

TITLE

Please revise your title according to PLOS Medicine's style. Your title must be nondeclarative and not a question. It should begin with main concept if possible. "Effect of" should be used only if causality can be inferred, i.e., for an RCT. Please place the study design ("A randomized controlled trial," "A retrospective study," "A modelling study," etc.) in the subtitle (ie, after a colon).

ETHICS

The ethical approval numbers from the ethical boards of the Chinese Academy of Medical Sciences between the protocol (CAMS & PUMC-IEC-2022-062) and the manuscript (CAMS&PUMC-IEC-2024-047) do not match. Please check and revise.

DATA AVAILABILITY

Thank you for providing a data availability statement. We feel that other researchers would not know how to request the data from the links you provide. Could you please provide a web or email address that provides a clearer way to access the data?

CODE AVAILABILITY

You have stated that the data code could be made available by contacting the responding author after approval. Could you please include this statement in your data availability statement?

ABSTRACT

1) Please clarify in the Abstract that the study is part of the Stroke Patients and Family Longitudinal Study in Rural China (SaFaRI, NCT05792618), a long-term passive (observational) follow-up of SINEMA cRCT participants and their spouses, as stated in your rebuttal. It should be clear to the reader that this is an observational study and not a pre-specified long-term outcome of the SINEMA trial.

2) l.72ff: Throughout the abstract and main text, please ensure that abbreviations, including statistical abbreviations, are defined the first time they are used (e.g. Q, CI etc.).

3) l.76/l.77: We suggest changing ‘cardiovascular mortality’ and ‘stroke mortality’ to ‘cardiovascular cause-specific mortality’ and ‘stroke cause-specific mortality’ (or similar). Please revise throughout the main text.

4) Since the primary objective of the SINEMA trial was to determine whether an integrated primary care-based mobile health intervention could improve stroke management, we feel that you should emphasize in the Abstract Conclusion that during six years of follow-up, no difference in stroke-related mortality was observed between the intervention group and the usual care group.

5) In the last sentence of the Abstract Methods and Findings section, please describe the main limitation(s) of the study's methodology.

6) Please ensure that all numbers presented in the abstract are present and identical to numbers presented in the main manuscript text.

7) Please include the important dependent variables that are adjusted for in the analyses.

8) Since you have included the ClinicalTrials.gov IDs below the Abstract, we suggest deleting the sentence on lines 68-70.

9) l.61: Please spell out ‘mHealth’ the first time used.

AUTHOR SUMMARY

1) l.110: Please remove ‘stroke mortality’ since the difference was not significant.

2) The last bullet point under ‘What did the researchers do and find?’ should be moved under last sub-heading.

3) We suggest changing the third bullet point under ‘What did the researchers do and find?’ to: “Stronger associations were observed among more vulnerable individuals, which may indicate the inequity-reducing impact of the intervention.”

4) We feel that the bullet points under the heading "What do these findings mean?" could be more concise. Please revise. Ideally, the subheading should contain 2-3 single sentence, concise bullet points that summarize the most important points of your study.

INTRODUCTION

If there has been a systematic review of the evidence related to your study (or you have conducted one), please refer to and reference that review and indicate whether it supports the need for your study.

METHODS AND RESULTS

1) Similar to comment 1 under "Abstract", please clarify that the manuscript describes the Stroke Patients and Family Longitudinal Study in Rural China (SaFaRI, NCT05792618), a long-term passive (observational) follow-up of SINEMA cRCT participants and their spouses.

2) l.168ff: “stroke survivors” - PLOS Medicine prefers the use of patient-centered language, e.g. “patients who were clinically stable after stroke” (or similar). Please revise throughout the manuscript.

3) l. 216: Please define ‘ID’ at first use.

4) ll.228-231, please change to: “For six decedents with missing dates of death (2% of confirmed deaths), dates were imputed using median survival times, which were 8.69 months for five individuals who died during the one-year trial phase and 46.39 months for one individual who died during the post-trial phase.”

5) l.277ff: Please ensure to introduce abbreviations the first time used (here: ‘HR’, introduced on line 281, but should be introduced on line 277).

6) l.283: “…while the fully adjusted model included these variables along with additional baseline variables noted to be significantly different between the arms at baseline (P<0.05) [16].” – we do not feel that providing a reference is sufficient. Please outline the variable adjusted for in the fully adjusted model or detail in the Appendix.

7) l.289: The terms gender and sex are not interchangeable (as discussed in https://www.who.int/health-topics/gender#tab=tab_1 ); please use the appropriate term.

8) l.335: The Table E title is ‘Baseline characteristics of decedents in the SINEMA trial by intervention arms at 70-months post-baseline’ – it is our understanding that you are presenting characteristics at 70 months post baseline, so the first "baseline" should be removed. Please revise the table title and text accordingly.

9) Figure 2: Please note that each figure should be self-explanatory on its own, i.e. we ask you to include a very brief description of what the intervention entailed in the figure description.

10) l.349ff: We feel that you should explain the results for stroke-specific mortality in more detail, because compared to the results for all-cause and cardiovascular-specific mortality, the results were not significant, i.e. there was no evidence of an association.

11) l.357: The reference to Table F appears to be incorrect. The results for the sub-distribution model are presented in Table G. Please carefully check that you have referenced the appropriate tables and figures throughout your manuscript and revise accordingly.

12) l.359: What about age, stroke duration, baseline systolic blood pressure and stroke type?

13) l.374: The term trend should be used only when the test for trend has been conducted. Please revise accordingly.

14) l.378: Please define ‘BMI’.

15) l.379: As noted in our comment 10), please carefully check that you have provided the relevant tables and have referred to the appropriate tables and figures throughout your manuscript. Table H does not show the results of the post-hoc analysis on changes in potential lifestyle factors.

16) Figure 1: Please define 'SD'. Please revise for patient-centered language.

17) Table 1: Please see the comment about the terms gender and sex and revise accordingly.

18) Tables: Please revise the reference callouts below the table to meet our formatting requirements.

DISCUSSION

General guidance: Please present and organize the Discussion as follows: a short, clear summary of the article's findings; what the study adds to existing research and where and why the results may differ from previous research; strengths and limitations of the study; implications and next steps for research, clinical practice, and/or public policy; one-paragraph conclusion.

1) l.388: “Our study is unique among the few cRCTs in LMICs…” – please clarify that the current study is of observational nature and a long-term follow-up of a RCT.

2) l.448: Please define ‘CVD’ at first use.

3) Please remove any subheadings including the Conclusion subheading.

REFERENCES

1) Please ensure that journal name abbreviations match those found in the National Center for Biotechnology Information (NCBI) databases (http://www.ncbi.nlm.nih.gov/nlmcatalog/journals), and are appropriately formatted and capitalized. For example, in reference [1], ‘The Lancet

Neurology’ should be ‘Lancet Neurol’.

2) Please also see https://journals.plos.org/plosmedicine/s/submission-guidelines#loc-references for further details on reference formatting.

SUPPLEMENTARY MATERIAL

1) Please ensure that all supplementary files are referenced in the main text.

2) In the published article, supporting information files are accessed only through a hyperlink attached to the captions. For this reason, you must list captions at the end of your manuscript file. You may include a caption within the supporting information file itself, as long as that caption is also provided in the manuscript file. Do not submit a separate caption file.

When SI files are contained with a single file:

Please label the file as ‘S1 Supporting Information’.

Please apply alphabetical labelling to each table and figure contained within the S1 file. For example, ‘Fig A’ to ‘Fig Z’ and ‘Table A’ to ‘Table Z’.

Plain text does not need to be labelled and can just be given a title as necessary. For example, ‘Statistical Analysis Plan’.

Please cite tables/figures as ‘Fig A in S1 Supporting Information’ and/or ‘Table A in S1 Supporting Information’, for example.

Please cite plain text as, ‘Statistical Analysis Plan in S1 Supporting Information’, for example.

When SI files are uploaded as separate files:

Please label tables as ‘S1 Table’ (so on) and figures as ‘S1 Fig’ (and so on).

Any additional documents (protocols/analysis plans etc.) can be labelled as ‘S1 Protocol’, for example. Please cite items as exactly as labelled.

General Editorial Requests

---

## [Editor Report · Decision Letter 3]

14 Feb 2025

Dear Dr. Yan,

Thank you very much for re-submitting your manuscript "Long-term mortality outcome of a primary care-based mobile health intervention for stroke management: six-year follow-up of a cluster-randomized controlled trial" (PMEDICINE-D-24-03550R3) for review by PLOS Medicine.

There are a few minor editorial issues that need to be addressed before we can accept the manuscript for publication; these are outlined at the end of this email. Please revise the paper accordingly, and submit the final revision within 1 week (Feb 21 2025).

Please ensure you address the specific points made by the editors. In your rebuttal letter you should indicate your response to the editors' comments and the changes you have made in the manuscript. Please submit a clean version of the paper as the main article file. A version with changes marked must also be uploaded as a marked up manuscript file. Please also check the guidelines for revised papers at http://journals.plos.org/plosmedicine/s/revising-your-manuscript for any that apply to your paper.

A reminder that when your manuscript is accepted, an uncorrected proof of your manuscript will be published online ahead of the final version, unless you've already opted out via the online submission form. If, for any reason, you do not want an earlier version of your manuscript published online or are unsure if you have already indicated as such, please let the journal staff know immediately at plosmedicine@plos.org.

If you have any questions in the meantime, please contact me directly at atosun@plos.org.

We look forward to receiving the revised manuscript.

Sincerely,

Alexandra Tosun, PhD

Associate Editor

PLOS Medicine

Requests from Editors:

1) Although we understand that the study is a long-term follow-up based on the intervention implemented as part of the SINEMA trial, the current study was not a pre-specified long-term outcome and is a passive observational study. Therefore, causality cannot be inferred. We recommend focusing on the discussion of the observed long-term and sustained differences between the intervention and control arms, rather than discussing the impact of the study, which would imply causality.

Please once again revise the manuscript carefully with regard to causal language. A few examples:

• l.381: “Over 66 months, the SINEMA intervention had led to a sustained…”

• l.412: “Our long-term follow-up study reported sustained impact…” (exchange ‘reported’ with ‘observed’)

• l.436: “…this sustained impact likely resulted”

2) Abstract, ll.54-56, please change to: “We investigated the association of a 12-month system-integrated technology-enabled model of care (SINEMA) intervention with mortality outcomes among patients experiencing stroke at six-years post-trial.”

3) The Data Availability Statement (DAS) requires revision. Please note that a study author cannot be the contact person for inquiries. Please provide an alternative contact.

4) Please once again revise for use of patient-centered language. Pleas note that person-centered language is constructed with the use of post-modified nouns (e.g. patients with stroke, patients experiencing stroke) putting the person first in the sentence structure. For example, ‘stroke patients’ should be changed to ‘patients experiencing stroke’ (or similar). Please carefully revise throughout.

---

## [Editor Report · Decision Letter 4]

21 Feb 2025

Dear Dr Yan, 

On behalf of my colleagues and the Academic Editor, Joshua Z Willey, I am pleased to inform you that we have agreed to publish your manuscript "Long-term mortality outcome of a primary care-based mobile health intervention for stroke management: six-year follow-up of a cluster-randomized controlled trial" (PMEDICINE-D-24-03550R4) in PLOS Medicine.

I appreciate your thorough responses to the reviewers' and editors' comments throughout the editorial process. We look forward to publishing your manuscript, and editorially there are only a few remaining minor stylistic points that should be addressed prior to publication. We will carefully check whether the changes have been made. If you have any questions or concerns regarding these final requests, please feel free to contact me at atosun@plos.org.

Please see below the minor points that we request you respond to:

1) Author Summary: Please change the second bullet point under ‘What did the researchers do and find?’ to: “Six years after the 12-month primary care-based mobile health intervention, we observed a reduction in all-cause and cardiovascular cause-specific mortality among patients with stroke in rural China, which may indicate sustained long-term benefits of the intervention.”

2) Author Summary: Please change the first bullet point under ‘What do these findings mean?’ to: “Our results extend previous within-trial evidence on the effectiveness of primary care-based strategies to promote secondary prevention of stroke in resource-limited settings by demonstrating a potential long-term association with mortality, even when the intervention was not requested further.”

Before your manuscript can be formally accepted you will need to complete some formatting changes, which you will receive in a follow up email (including the editorial points above). Please be aware that it may take several days for you to receive this email; during this time no action is required by you. Once you have received these formatting requests, please note that your manuscript will not be scheduled for publication until you have made the required changes.

PRESS

Sincerely, 

Alexandra Tosun, PhD 

Associate Editor 

PLOS Medicine